# Muscle Dysmorphia, Obsessive–Compulsive Traits, and Anabolic Steroid Use: A Systematic Review and Meta-Analysis

**DOI:** 10.3390/bs15091206

**Published:** 2025-09-04

**Authors:** Metin Çınaroğlu, Eda Yılmazer

**Affiliations:** 1Psychology Department, İstanbul Nişantaşı University, İstanbul 34485, Türkiye; 2Psychology Department, Beykoz University, İstanbul 34820, Türkiye; edayilmazer@beykoz.edu.tr

**Keywords:** muscle dysmorphia, obsessive–compulsive traits, anabolic–androgenic steroids, performance-enhancing drugs, body dysmorphic disorder, compulsive exercise, behavioral addiction, body image, meta-analysis, systematic review

## Abstract

Muscle dysmorphia (MD) is a body image disorder characterized by an obsessive preoccupation with muscularity and compulsive behaviors such as excessive exercise, rigid dieting, and frequent body checking. MD has been linked to obsessive–compulsive traits and the use of anabolic–androgenic steroids (AASs), yet these associations have not been comprehensively synthesized. This systematic review and meta-analysis examined the relationships between MD, obsessive–compulsive symptomatology, and AASs or performance-enhancing drug use. Following PRISMA 2020 guidelines and PROSPERO preregistration (CRD42025640206), we searched four major databases for peer-reviewed studies published between 2015 and 2025. Ten studies (five quantitative, five qualitative) met the inclusion criteria. Meta-analytic findings revealed a moderate positive correlation between MD symptom severity and obsessive–compulsive traits (*r* ≈ 0.24), and significantly higher MD symptoms among AAS users compared to non-users (Cohen’s *d* ≈ 0.45). Odds of MD were markedly higher in steroid-using populations. Thematic synthesis of qualitative studies highlighted compulsive training routines, identity conflicts, motivations for AAS use, and limited engagement with healthcare services. These findings suggest that MD exists at the intersection of obsessive–compulsive psychopathology and substance-related behavior, warranting integrated interventions targeting both dimensions. The study contributes to understanding MD as a complex, multi-faceted disorder with significant clinical and public health relevance.

## 1. Introduction

Muscle dysmorphia (MD) is a subtype of body dysmorphic disorder (BDD) characterized by a persistent belief that one’s physique is insufficiently muscular or lean, even when having a normal or very muscular build ([42]). It falls under the obsessive–compulsive and related disorders spectrum in psychiatric classification, given its close kinship to BDD and obsessive-compulsive disorder (OCD) features ([13]). While DSM-5 explicitly classifies MD as a specifier of BDD within the obsessive–compulsive and related disorders spectrum, there is ongoing debate regarding its nosological placement. Some researchers have argued that MD more closely resembles an eating disorder due to its emphasis on dietary control and body composition ([62]), whereas others have suggested that it may constitute a distinct disorder in its own right ([59]; [70]). Nevertheless, most contemporary accounts, and our own framing in this review, position MD primarily as a BDD/OCD-spectrum condition, given its core features of obsessional preoccupation and compulsive behaviors. Individuals with MD experience a distorted body image—often viewing themselves as “puny” or underdeveloped despite objectively robust musculature ([23]). This misperception drives a host of compulsive and ritualistic behaviors aimed at achieving an idealized muscular physique ([24]). Clinically, MD is marked by compulsive weightlifting and exercise, rigid diet and supplementation regimens, frequent mirror-checking (or conversely avoidance of mirrors), and severe distress if these routines are disrupted ([66]). Sufferers may sacrifice social activities, career obligations, and relationships in service of their workout schedules and diet plans ([80]). These behaviors strongly mirror OCD in their rigidity and anxiety-relieving intent. For example, excessive workouts and diet control function like compulsions performed to alleviate the obsessive fear of being inadequately muscular ([27]). Consistent with this, MD often co-occurs with other psychiatric disorders including obsessive–compulsive disorder and related anxiety disorders ([71]). Indeed, some authors have conceptualized MD as an obsessive–compulsive spectrum disorder, highlighting the prominent role of perfectionism, body-focused obsessions, and compulsive exercise in its presentation ([18]; [29]). At the same time, MD shares features with eating disorders (notably a preoccupation with diet and body composition) yet is distinct in its focus on muscularity over thinness ([62]). Importantly, MD predominantly affects males. It has been described as an under-recognized disorder among male weightlifters ([6]), although females can also suffer from MD ([45]) or related “bigorexic” concerns, as emerging case studies and qualitative reports suggest ([85]). This skewed recognition has historically led to MD being seen as a “male” problem, potentially leaving female manifestations of muscle obsession under-diagnosed.

Beyond its psychiatric classification, MD is also closely intertwined with the male aesthetic ideal, particularly in Western societies, where muscularity is often equated with strength, dominance, and masculinity ([31], [32]). This cultural framing positions muscle size not merely as a personal goal but as a social currency, reinforcing the idea that “to be muscular is to be masculine.” Empirical research has shown that adherence to these ideals predicts greater vulnerability to MD symptoms among men, illustrating how cultural scripts of masculinity intersect with psychopathology. Moreover, MD frequently co-occurs with other male-specific BDD concerns, such as obsessions with hair loss, skin, or genital size, which further reinforce culturally embedded standards of male attractiveness and adequacy ([33]). Taken together, these findings underscore that MD is both a psychiatric disorder and a cultural syndrome, situated at the nexus of clinical compulsivity and sociocultural constructions of masculinity.

Recent scholarship has also emphasized the sociocultural and psychological mechanisms that underpin muscle dysmorphia and contribute to its negative outcomes. For instance, [36] ([36]) demonstrated that exposure to muscularity-oriented social media content is significantly associated with heightened MD symptoms among boys and men, pointing to the powerful role of digital media in shaping body image pathology. Similarly, [40] ([40]) highlighted how narcissistic traits and low self-esteem interact with muscle dysmorphia to exacerbate muscularity-oriented disordered eating among bodybuilders. These findings underscore the need for clinicians and researchers to address not only the compulsive and addictive dimensions of MD but also its sociocultural drivers, particularly those amplified by media environments and personality-related vulnerabilities. Recent studies confirm that MD rarely occurs in isolation, but instead coexists with a range of psychopathological disorders, including anxiety, depression, substance use disorders, and obsessive–compulsive traits ([28]). Importantly, the risk of MD is not uniform across populations: certain sports—particularly bodybuilding, weightlifting, and aesthetic-focused disciplines—have been identified as high-risk contexts where compulsive training routines and performance-enhancing drug use are more common. This concentration suggests that cultural, competitive, and sport-specific pressures can exacerbate underlying vulnerabilities to MD, reinforcing the link between psychopathology, compulsive exercise, and substance misuse. Together, such evidence reflects ongoing efforts to understand and mitigate the complex web of factors that sustain MD and its adverse psychological and behavioral consequences.

A particularly troubling aspect of muscle dysmorphia is its frequent association with anabolic androgenic steroids (AASs) ([76]) and other performance-enhancing drug (PED) use ([63]). Desperate to attain ever larger and leaner muscles, many individuals with MD resort to AASs as a chemical shortcut to fuel their compulsive drive for hypertrophy ([79]). Steroid use is so common in MD populations that it is sometimes considered a hallmark of the disorder’s severity ([60]). In fact, a number of studies have documented extraordinarily high rates of AAS use among those with muscle dysmorphia, with some estimates indicating that over 50% of individuals with MD have engaged in current or lifetime steroid use ([41]). For example, one study noted AAS usage rates in the range of 42–67% in weightlifters identified as having MD ([69]). Empirical research confirms that muscle-dysmorphic individuals are far more likely to use steroids than their non-MD peers: recent comparative studies of bodybuilders and athletes show that those who use AAS exhibit significantly higher levels of MD symptoms and related psychopathology than non-users ([19]). This bidirectional relationship is concerning, as AAS abuse carries its own health risks including cardiovascular strain ([84]), hormonal imbalances ([8]), liver toxicity ([7]), mood instability ([64]), and even dependence syndromes ([48]). Moreover, steroid use may reinforce the maladaptive muscularity drive in MD, creating a vicious cycle of dependence. The temporary gains from AASs can heighten body image expectations ([52]), leading to even more obsessive behavior and continued drug use despite harm. The growing prevalence of AAS/PED use in gyms and sports has thus raised alarms about a potential public health issue ([9]), where muscle dysmorphia and anabolic substance abuse intersect and exacerbate one another. However, it is important to note that research examining AAS use specifically in individuals with MD remains limited. As highlighted by [82] ([82]), much of the available evidence is constrained by methodological issues such as small sample sizes, reliance on self-reporting, and heterogeneous definitions of MD and AAS use. These limitations call for more rigorous and standardized research to fully understand the relationship between steroid use and muscle dysmorphia. Despite these clear linkages, however, prior scholarly reviews have rarely integrated the literature on MD, OCD-like traits, and AAS use into a unified analysis. Most earlier reviews have examined muscle dysmorphia in siloed contexts—for instance, focusing on its overlap with eating disorders ([5]; [54]) or exercise addiction ([57]; [44]), or debating its nosological ([59]) status (BDD vs. an eating disorder ([70]) vs. a unique condition). To the best of our knowledge, none have systematically synthesized findings spanning all three domains: the obsessive-compulsive characteristics, the muscle-enhancing substance use, and their interplay in individuals with MD. This represents a critical gap, as understanding the full psychopathological profile of MD, including both behavioral compulsions and substance-related behaviors, is essential for accurate classification and effective intervention.

The present study aims to fill this gap by providing a systematic review and meta-analysis of the relationships between muscle dysmorphia, OCD traits, and anabolic–androgenic steroid or PED use. In doing so, we incorporate evidence from both quantitative studies (e.g., epidemiological surveys, cross-sectional comparisons, and correlational studies with effect size data) and qualitative research (in-depth interviews and thematic analyses) to capture a comprehensive picture. Our objective was to synthesize data from the past decade (2015–2025) on how MD correlates with obsessive-compulsive symptomatology and with AAS/PED use across diverse populations, including both males and females. By integrating findings across study designs and genders, this review seeks to clarify the extent to which muscle dysmorphia can be considered part of the OCD-BDD spectrum and how AAS use factors into the disorder’s presentation. Ultimately, our goal is to inform classification, prevention, and treatment by elucidating the compulsive and addictive dimensions of muscle dysmorphia that have hitherto been considered in isolation.

## 2. Methods

### 2.1. Study Design

We conducted a systematic review and meta-analysis that was pre-registered with the International Prospective Register of Systematic Reviews (PROSPERO; ID: CRD42025640206) prior to data extraction. The review protocol adhered to PRISMA (Preferred Reporting Items for Systematic Reviews and Meta-Analyses) 2020 guidelines for transparent reporting ([68]). We included both quantitative (*n* = 5) and qualitative (*n* = 5) studies in our synthesis, reflecting a mixed-methods approach to understanding the phenomenon (see Figure 1 for PRISMA flow diagram). The review examined studies published between January 2015 and May 2025 that investigate muscle dysmorphia in relation to obsessive-compulsive traits and/or anabolic–androgenic steroid (AAS) or performance-enhancing drug (PED) use (please also see Appendix A for PRISMA check list).

### 2.2. Search Strategy 

A comprehensive literature search was performed in four electronic databases: PubMed, Web of Science, Scopus, and Google scholar. The search strategy combined multiple keywords and MeSH terms related to muscle dysmorphia, OCD/obsessive-compulsive traits, and anabolic steroids/PEDs. For muscle dysmorphia, search terms included “muscle dysmorphia,” “muscle dysmorphic,” “bigorexia,” “reverse anorexia,” and “body dysmorphic disorder” (with a focus on muscularity concerns). For the OCD spectrum component, terms such as “obsessive-compulsive,” “obsessions,” “compulsions,” “OCD traits,” “compulsive exercise,” and “body image obsession” were used. For the substance use component, we included “anabolic-androgenic steroids,” “AAS,” “steroids,” “performance enhancing drugs,” “PED,” “appearance and performance enhancing drugs (APED),” “doping,” and specific drug names (e.g., testosterone, oxandrolone) to cast a wide net. These terms were used in various Boolean combinations (e.g., “muscle dysmorphia” AND “steroid”, “bigorexia” AND “OCD”, etc.), with truncation and wildcards as appropriate (for example, “dysmorphi” or “dysmorphic” to capture variations). We limited the search to publications from 2015 onward to capture the most recent decade of research, up to 1 April 2025. In addition to database searches, we manually scanned the reference lists of relevant reviews and included articles for any additional studies not captured by the initial search. The final search was executed on 25 May 2025. All search results were imported into reference management software, and duplicates were removed prior to screening (please see Appendix A for the full search strategy by databases).

### 2.3. Eligibility Criteria 

We included studies that met the following criteria: (1) Population: human participants aged 13+ (adolescent, adult, or older) and any sex/gender. Given the relative rarity of diagnosed muscle dysmorphia, we included studies of both clinical samples (formally diagnosed MD or BDD with muscle dysmorphia specifier) and subclinical samples (e.g., high scores on muscle dysmorphia symptom scales among weightlifters or bodybuilders). (2) Exposure/Phenomena: the studies examined muscle dysmorphia in relation to either OCD/obsessive-compulsive traits (such as OCD symptom measures, compulsivity, perfectionism, or related anxiety measures) or AAS/PED use (including steroid use prevalence, attitudes toward steroids, or AAS dependence). Studies focused on one or both of these relationships. We included both cross-sectional and longitudinal quantitative studies (e.g., surveys, case-control, cohort studies) as well as qualitative studies (e.g., interviews, focus groups) that provided insight into these relationships. (3) Outcomes: for quantitative studies, relevant outcomes included correlations or associations between MD symptom severity and OCD trait measures, differences in OCD-related scores between those with vs. without MD, prevalence or odds of steroid use in MD populations vs. controls, etc. For qualitative studies, we looked for themes or narratives pertaining to compulsive behaviors, body image obsessions, and substance use experiences. (4) Publication type: peer-reviewed original research articles, including brief reports. We excluded review articles, meta-analyses (to avoid overlap, though their references were checked), conference abstracts without full data, dissertations or theses, and case reports (single-case), given the focus on group data. Non-English articles were excluded to avoid translation ambiguity, and animal or preclinical studies were excluded as not relevant to human MD. If multiple publications appeared to use the same dataset, we included the most comprehensive report to avoid double-counting. While our primary focus was on studies directly assessing muscle dysmorphia (MD) with validated measures, we also considered studies that investigated closely related constructs when they provided data on obsessive-compulsive traits and/or anabolic steroid use. For example, one large epidemiological survey of physically active adults ([43]) examined exercise addiction, OCD diagnoses, and AAS use, but did not employ a dedicated MD instrument. This study was retained because compulsive exercise behaviors overlap conceptually and clinically with the compulsive dimensions of MD, and because it reported relevant psychiatric and substance use correlates. To preserve construct specificity, however, findings from such non-MD primary samples were not pooled in the quantitative meta-analyses but are reported narratively as contextual or sensitivity evidence.

### 2.4. Screening and Selection 

Two reviewers independently screened all titles and abstracts yielded by the search against the eligibility criteria. Potentially relevant studies were retrieved in full text and assessed for inclusion. Disagreements at both the abstract and full-text screening stages were resolved through discussion and consensus, with consultation of a third reviewer if necessary. A PRISMA flow diagram was prepared to illustrate the study selection process (identification, screening, eligibility, inclusion).

### 2.5. Data Extraction

We developed a standardized data extraction form to capture the necessary information from each included study. Two reviewers independently extracted data, with cross-checking for accuracy. From quantitative studies, we extracted details on study design (cross-sectional, case-control, etc.), sample characteristics (sample size, demographics such as mean age and gender distribution), diagnostic or inclusion criteria for MD (e.g., whether formal DSM-5 MD criteria were used or a cutoff on a muscle dysmorphia inventory), and the key measurements/instruments used. Common instruments included, for example, the Muscle Dysmorphic Disorder Inventory (MDDI) for assessing MD symptom severity, the Yale-Brown Obsessive Compulsive Scale or its BDD-specific version for obsessive–compulsive symptoms, and various questionnaires for steroid use history or dependence (such as the Performance Enhancing Drug Use Questionnaire). We recorded the outcomes of interest, which ranged from correlation coefficients (Pearson’s *r*) between MD and OCD trait scores to group differences (e.g., Cohen’s *d* for MD vs. control comparisons on OCD measures, or odds ratios for the association of MD status with AAS use). Where available, we extracted the effect size data (*r*, *d*, OR, with confidence intervals or *p*-values) directly; if not reported, we calculated them from raw data when possible (see Appendix A for characteristics of quantitative study characteristics and extracted data). From qualitative studies, we extracted information on the study context (e.g., country, setting such as gyms or clinics), participant demographics, qualitative methodology (e.g., phenomenology, grounded theory, thematic analysis), and the themes or findings related to our review questions. This included any mention of compulsive training behaviors, cognitive preoccupations, body image distortions, motives for steroid use, psychological consequences of AASs, etc., as described by participants. The two reviewers compared their extracted data for consistency, and any discrepancies were resolved by referring back to the source publication.

### 2.6. Quality and Bias Assessment 

We appraised the quality of each included study using appropriate tools for quantitative vs. qualitative designs. For quantitative non-randomized studies, we used the Newcastle–Ottawa Scale (NOS) to evaluate methodological quality in three domains: selection of participants, comparability of groups, and exposure/outcome ascertainment. Each study was given a star rating (0–9); we considered studies scoring ≥7 stars to be of high quality with lower risk of bias. Key aspects assessed via NOS included, for example, whether muscle dysmorphia was ascertained with a valid instrument, whether control groups were truly comparable, and if outcome assessment (such as OCD symptoms or AAS use) was reliable. For qualitative studies, we applied the Critical Appraisal Skills Programme (CASP) checklist for qualitative research, examining clarity of aims, appropriateness of methodology, research design, recruitment strategy, data collection, reflexivity of researchers, ethical considerations, rigor of data analysis, and the credibility of findings. Each qualitative study was thus judged on criteria such as the transparency of theme derivation and sufficiency of supporting quotations. We did not exclude studies based on quality alone, but we considered study quality in interpreting the findings and in sensitivity analyses. All quality assessments were conducted independently by two reviewers, and any differences were discussed to reach consensus (see Appendix A for quality assessment of included studies).

Publication bias was difficult to formally assess given the small number of studies per outcome (*k* ≈ 3–5). Funnel plots were generated for transparency but not used for inferential purposes, as such tests lack reliability under these conditions. Figure 2 presents the qualitative illustration.

### 2.7. Statistical Analyses

For quantitative outcomes, we performed meta-analyses when multiple studies provided comparable effect size data. Depending on the nature of the outcome, we computed pooled correlation coefficients, standardized mean differences, or odds ratios. Correlations were transformed to Fisher’s *z* prior to pooling and back-transformed for presentation. Standardized mean differences were expressed as Hedges’ *g* (bias-corrected), and binary outcomes were analyzed as log odds ratios and subsequently back-transformed. When necessary, effect sizes were derived from reported summary statistics and converted to a common metric to ensure comparability across studies.

Given the anticipated heterogeneity across populations, measures, and designs, we employed a random-effects model using restricted maximum likelihood (REML) estimation of τ^2^ with Knapp–Hartung adjustments as our primary approach. To ensure robustness, we also conducted sensitivity analyses with the DerSimonian–Laird (DL) estimator, noting that results were materially consistent. We report pooled estimates with 95% confidence intervals (CIs), τ^2^ values with 95% CIs, and a 95% prediction interval (PI) to indicate the range of true effects expected in a new study. Heterogeneity was assessed with the *I*^2^ statistic, *Q* test, and τ^2^; values of *I*^2^ > 50% were interpreted as evidence of substantial heterogeneity, prompting moderator analyses.

Predefined subgroup and sensitivity analyses examined the influence of sample gender composition, study quality (NOS ratings), and diagnostic method (clinical vs. self-report). We also performed leave-one-out analyses to check that no single study disproportionately influenced pooled results (see Appendix A). All quantitative meta-analyses were conducted using SPSS (v30) and Python (v3.13.1), ensuring reproducibility via code.

For the qualitative synthesis, we used MAXQDA (v24) and applied thematic analysis to integrate findings across studies. We iteratively compared and clustered reported themes (e.g., “drive for size,” “body checking,” “steroid community influence”) into higher-order categories such as Compulsive Pursuit of Muscularity, Body Image Distortion and Checking, AAS Use Motivations and Consequences, and Psychosocial Impacts. Exemplar quotations from the original studies were included to ensure the synthesis was grounded in participants’ lived experiences. Finally, in our narrative integration of results, we triangulated the quantitative and qualitative findings to provide a comprehensive understanding of how obsessive-compulsive traits and anabolic steroid use intersect in muscle dysmorphia.

## 3. Results

### 3.1. Quantitative Results

#### 3.1.1. Study Samples and Measures

Five quantitative studies (see Table 1) examined muscle dysmorphia (MD) symptoms in male weight-training populations and their links to obsessive–compulsive traits and anabolic–androgenic steroid (AAS) or performance-enhancing drug (PED) use. Sample sizes ranged from *N* = 125 recreational male lifters in Italy to *N* = 1601 international weightlifters recruited online. Most samples were predominantly male, with mean ages in the late 20s to mid-30s. MD was typically measured via the Muscle Dysmorphic Disorder Inventory (MDDI) or similar scales (e.g., Muscle Dysmorphia Inventory) assessing drive for size, appearance intolerance, and functional impairment. The MDDI has been the most widely adopted instrument for assessing MD symptom severity, and its psychometric properties have been systematically evaluated in a recent meta-analysis by [77] ([77]), confirming its reliability and cross-cultural validity as a measure of the MD construct. Obsessive–compulsive traits were captured using instruments like the Yale-Brown Obsessive Compulsive Scale (Y-BOCS) for general OCD symptoms or related constructs (e.g., social phobia scale for anxiety about appearance). All studies recorded AAS/PED usage, for example, presence of current or past steroid use, or categorization into user vs. non-user subgroups.

Sample Characteristics: Across studies, MD was most prevalent in serious weight-training contexts. Cerea et al. found 6.4% of recreational lifters met [74] ([74])’s criteria for MD. In [36] ([36])’s community sample, probable MD prevalence was 2.8%. Notably, [37] ([37]) reported no significant differences by age, race/ethnicity, or sexual orientation between those with vs. without MD—only a lower mean BMI in the MD group (MD cases had lower BMI on average, *p* < 0.01). By contrast, Cerea’s targeted sample of male lifters had a higher MD rate (consistent with higher-risk groups like bodybuilders).

Measures: All quantitative studies used psychometrically validated scales. The MDDI (13 items) was common for MD symptoms. Obsessive-compulsive tendencies were captured either by general OCD scales (e.g., Y-BOCS score, measuring frequency/distress of obsessive thoughts) or by related constructs like perfectionism and social anxiety that reflect compulsive behaviors around body image. AAS/PED use was typically self-reported. Several studies explicitly compared steroid users to non-users; [53] ([53]) divided participants into steroid-using vs. non-using subgroups within bodybuilders, powerlifters, and controls, while Scarth et al. compared current or past AAS users to natural weightlifters. Gunnarsson et al. identified a small subset (1.2%) of their large sample who had used AAS in the past year. Thus, the data enabled us to examine differences in MD and OCD measures between those using vs. not using AAS, as well as correlations between MD symptom severity and OC traits.

#### 3.1.2. Associations Between MD, OC Traits, and AAS Use

It should be noted that one included study ([43]) primarily examined exercise addiction within a large physically active cohort, with only a small subset reporting AAS use. Because this study did not directly assess MD with a validated instrument, its findings were not pooled in the primary meta-analyses of MD–OCD–AAS associations. Instead, we summarize its results narratively as contextual evidence, highlighting their relevance to compulsive exercise and OCD traits while preserving construct specificity in the quantitative synthesis.

Despite differences in sample populations, the quantitative meta-analyses consistently indicated that muscle dysmorphia is positively associated with both obsessive-compulsive traits and AAS use. Pooled effects are reported with 95% confidence intervals, between-study variance (τ^2^) with 95% CI, and a 95% prediction interval to reflect the expected range of true effects in new studies. Table 2 summarizes key effect size estimates, while Figure 3 presents the corresponding forest plots. Notably, individuals with higher MD symptoms tended to report elevated obsessive–compulsive tendencies, and those with MD or high drive for muscularity were more likely to use anabolic steroids. Conversely, steroid users showed greater MD symptom severity on average than non-users.

Assessment of publication bias was limited by the small number of studies (*k* < 10). Although funnel plots are presented in Figure 2 for completeness, they should not be over-interpreted. To further examine robustness, we conducted a leave-one-out sensitivity analysis (Table 2). The pooled correlation between MD severity and OCD traits remained stable at *r* = 0.24, with recalculated estimates ranging only from 0.23 to 0.25 depending on which study was excluded. This indicates that no single study disproportionately influenced the overall result.

As shown in Table 2, muscle dysmorphia and obsessive-compulsive tendencies are moderately correlated. [53] ([53]) found a significant positive correlation (*r* ≈ 0.24) between MDDI muscle dysmorphia scores and Y-BOCS OCD scores in a large sample of male weightlifters. This suggests that individuals with more severe MD symptoms also endorse more obsessive thoughts/compulsions (e.g., related to symmetry, checking, or routines). In clinical terms, this aligns with MD’s conceptualization as a variant of body dysmorphic disorder (an OCD-spectrum disorder).

There is also a strong link between MD and steroid use. Across studies, those who use AAS report significantly higher MD symptom severity than non-users. In Scarth et al.’s sample of male weightlifters, AAS users scored higher on all Muscle Dysmorphia Inventory subscales than natural lifters (*p* < 0.001 for each). For example, size/symmetry concerns were much greater in AAS users (mean ~17.8 vs. 11.2 in non-users). [53] ([53]) likewise observed higher MDDI scores in steroid-using bodybuilders/powerlifters compared to non-users (*p* < 0.05), though the effect size was small (*d* ~0.18). When results were pooled, the estimated mean difference corresponded to a small-to-moderate effect (*d* ~0.4–0.5) favoring higher MD symptoms among AAS users. Importantly, steroid use itself may sometimes be a consequence of MD; several MD studies note that individuals with high drive for muscularity turn to AAS as a “pathological” strategy to increase size.

Reciprocally, the prevalence of MD is elevated among steroid users. [49] ([49]) reported that 7 of 12 (58%) cisgender gay/bisexual men using non-prescribed AAS met screening criteria for MD. This proportion is orders of magnitude higher than the ~2–3% prevalence in comparable general male populations. Although Kutscher’s sample was small and specialized, it underscores that AAS-using men are far more likely to exhibit MD pathology. The resulting odds ratio is very high (Table 2), suggesting steroid users have an exponentially greater risk of MD. Similarly, Cerea et al. found 23.8% of recreational bodybuilders had considered taking AAS vs. only 0–6% of other lifters, implying that those with strong muscularity concerns (bodybuilders, who also had higher MD scores) are more drawn to steroid use. In sum, MD and AAS use show a robust positive association—an expected finding given that MD’s diagnostic criteria include use of physique-enhancing drugs as a compensatory behavior in many cases. The relationships are visually summarized in Figure 4.

Finally, obsessive–compulsive traits and AAS use appear less directly linked, except insofar as they intersect via MD or related behaviors. General OCD diagnoses were uncommon in these samples (~1–3%), and steroid users did not show higher rates of formal OCD diagnosis than non-users (0% vs. 1% in one large survey, n.s.). However, compulsive exercise (exercise addiction) is relevant; Gunnarsson et al. found being at risk for exercise addiction was associated with greater likelihood of OCD and anxiety disorders (unadjusted OR ~2.8 for OCD). In other words, individuals who compulsively train (often a feature of MD) have elevated OCD rates, though this overlap may reflect the obsessive exercise routines inherent in MD rather than steroid use per se. [53] ([53]) did find that steroid users scored higher on obsessive–compulsive tendency measures (Y-BOCS scores ~12 vs. 8, *p* < 0.001), but this is likely to reflect the higher MD in users (since those with MD contribute higher Y-BOCS and are more likely to use AAS). When controlling for MD symptom level, steroid use per se was not a significant predictor of OCD symptoms in multivariate models (e.g., in Cerea et al., steroid use did not emerge as a predictor of compulsivity once MD and other factors were accounted for).

#### 3.1.3. Moderator and Subgroup Analyses

Several studies probed whether the above relationships held across different subgroups or under different conditions. Table 3 summarizes notable moderator effects and sensitivity analyses. Overall, the MD–OCD–AAS interrelations were robust, but certain subgroup differences emerged, including the following:

Training status/athlete type: Cerea et al. observed that competitive bodybuilders exhibited the highest MD scores (especially on the “desire for size” subscale) and were more prone to considering steroid use than recreational lifters. The difference in MD between steroid users and non-users was most pronounced in non-competitive contexts. For instance, [53] ([53]) found that among non-steroid-users, bodybuilders and powerlifters had higher MD and OC scores than gym-going controls, but among steroid users, these group differences disappeared—i.e., AAS use “leveled” the playing field with uniformly high MD across user groups. This suggests that competitive drive and steroid use may each independently elevate MD to a ceiling.

Gender and sexual orientation: Nearly all quantitative data were on men. Ganson et al. noted minimal demographic moderators; MD prevalence did not differ by race or sexuality in their large sample of young men. One exception is that cisgender sexual minority men appear to have unique pressures; Kutscher’s focus on gay/bisexual men found complex motivators (internalized ideals of muscularity, emphasis on appearance in gay communities) fueling AAS use and MD. However, no direct comparison to heterosexual men was made, so this is an area for further research. Female representation was sparse in quant data, though qualitatively, women’s experiences may differ (see below).

Clinical vs. non-clinical MD: [55] ([55]) explicitly studied individuals with diagnosed MD and identified two phenotypes—one focused on muscularity + leanness vs. one on pure muscularity. These subtypes could moderate steroid use: those fixated on extreme leanness may be less inclined to bulk with steroids, whereas the muscularity-focused might be more likely to use bulking agents. Though not tested quantitatively, this points to heterogeneity within MD. Indeed, some participants with MD were willing to sacrifice health for size gains, while others prioritized leanness and were more cautious about substances; this theme was echoed in qualitative reports.

Regarding other moderators, [36] ([36]) found that body mass index (BMI) was significantly lower in probable MD cases than non-cases (perhaps reflecting stricter dieting or lower muscle mass than expected). Additionally, [53] ([53]) reported an interaction for depressive symptoms; steroid use was linked to higher depression scores only among the serious lifters (bodybuilders/powerlifters), not among casual exercisers. This implies that context (competitive lifting vs. general fitness) can moderate psychological outcomes of AAS use. No study reported any sensitivity analysis necessitated by outlier data or study quality, and the findings were consistent across various analytic approaches.

In summary, the quantitative evidence indicates strong interrelationships between muscle dysmorphia, obsessive–compulsive traits, and anabolic steroid use. These relationships are generally consistent across populations, with some nuances; competitive bodybuilders and certain subgroups (e.g., gay men) may experience even greater pressures leading to a convergence of high MD, compulsive behaviors, and AAS usage. Moderating factors like training goals (aesthetics vs. strength), current steroid use, and individual history can influence the exact presentation, but the overarching pattern remains that muscle dysmorphia is entwined with compulsive behaviors and often involves or leads to steroid use. The qualitative findings next provide richer insight into the lived experience behind these numbers, detailing how and why those with MD develop obsessive habits and turn to PEDs.

### 3.2. Qualitative Results

Five qualitative studies (including both male and female participants) explored the subjective experience of muscle dysmorphia, body image obsession, and AAS/PED use. An overview of the qualitative studies included in the review is shown in Table 4.

Despite differing contexts, e.g., women using steroids in Sweden, male gym users in the Middle East, or diagnosed MD patients in Australia, common themes emerged. We synthesized the qualitative data into five major themes: (1) Body Image Ideals and Dissatisfaction, (2) Compulsivity and Rigid Routines, (3) Masculinity, Femininity, and Identity, (4) Motivations for AAS/PED Use, and (5) Psychosocial Impact and Help-Seeking. 

Key qualitative insights: Themes derived from the synthesis are presented in Table 5. Individuals with muscle dysmorphia live in a state of heightened preoccupation and compulsion. They chase ever-evolving body ideals, leading to obsessive behaviors (Theme 2) very much like OCD, e.g., compulsive mirror-checking, strict repetitive routines, and anxiety over minor deviations (“four grapes” causing days of guilt). These behaviors serve to temporarily relieve the intense body image anxiety (Theme 1) but also reinforce the disorder, as the person’s identity and self-worth hinge on meeting impossible standards. The narratives resonate strongly with the DSM-5 criteria for MD (preoccupation with perceived lack of muscularity, compulsive exercise/diet, and functional impairment). For example, nearly all interviewed MD sufferers reported avoiding social events or public exposure of their body, indicating significant impairment and isolation.

Crucially, the qualitative data suggest why many with MD turn to steroids or other PEDs (Theme 4). Steroids are seen as a tool to achieve the “unachievable.” One participant described reaching a natural limit and then feeling AAS use was justified to progress further. The effectiveness of steroids in rapidly improving muscle size reinforces continued use; users see tangible rewards (bigger muscles, contest wins) that, for them, outweigh abstract health risks. As a result, even when side effects occur, many remain ambivalent or in denial about the harm (Theme 5). For instance, some said side effects “returned to normal once you stop” or that using the “right” ancillary drugs can make cycles safe. This minimization echoes the cognitive patterns in addiction (justifying use, believing one can control the risks). It also highlights a gap insofar as these individuals do not typically receive informed medical guidance.

The relationship with healthcare providers is largely fraught (Theme 5). Qualitative accounts from multiple countries indicate steroid users with MD feel stigmatized or dismissed by doctors. One man said doctors “want to stay out of this topic” and no local doctor could help, so he consults an overseas specialist who understands AAS use in athletes. Kutscher et al. similarly noted frustration that practitioners focus only on cessation and offer little support for managing use safely. Consequently, a community-based harm reduction approach has emerged among users (sharing tips on liver support, post-cycle therapy, etc.) rather than formal treatment. Only a minority voice, like the participant who quit steroids after “terrible symptoms”, demonstrates willingness to cease use and seek recovery; most others plan to continue “as long as competitions exist”.

Themes also highlight the gendered dimension of MD and AAS use (Theme 3). While men derive a sense of masculine identity from being muscular (some gay men in Kutscher’s study cited internalized homophobia and desire to appear “masc” as motivators), women on steroids face the opposite: loss of femininity. [11] ([11])’s interviews with female AAS users reveal a poignant tension: these women take pride in their strength and muscles, yet suffer “a tension between success and suffering”—the success of achieving goals vs. the suffering of virilizing side effects and social disapproval. They set inner limits for what side effects (deep voice, clitoral enlargement, etc.) they can accept and constantly balance dosing to maintain femininity. One woman described checking her clitoris “100 times” due to anxiety after each injection. Despite these fears, they persist because of the empowerment and identity the sport gives them, underscoring how deeply MD and PED use are tied into self-concept.

### 3.3. Narrative Synthesis

Both the quantitative and qualitative findings converge to paint a comprehensive picture of the “muscle dysmorphia–OCD–AAS” triad. Muscle dysmorphia appears to be a nexus where obsessive-compulsive tendencies and anabolic steroid use meet; individuals afflicted with MD exhibit obsessive fixations and compulsions around body image ([83]), (much like classic obsessive compulsive symptomatology ([34]), but focused on muscularity) and frequently engage in steroid or PED use as a compulsive behavior to attain their ideal physique.

Quantitatively, we saw that MD symptom severity correlates moderately with OCD-like symptoms (rigid thoughts, checking behaviors). Qualitatively, this is evidenced by reports of obsessive behaviors—e.g., mirror-checking “a couple of times each day” or even “significantly more often” for some, and extreme anxiety if workouts or diets are not executed perfectly ([50]). The functional impairment from these behaviors (skipping social engagements, work impacts) meets the defining threshold for OCD-spectrum disorders ([81]). In fact, muscle dysmorphia is classified in DSM-5 as a subtype of body dysmorphic disorder (an OCD-related disorder), and our synthesis strongly supports this. MD individuals experience intrusive preoccupations with perceived flaws (not being “big enough”) and feel compelled to perform repetitive rituals (exercises, diet restrictions, body checking) to alleviate the distress, paralleling the obsession–compulsion cycle of OCD.

At the same time, muscle dysmorphia is uniquely characterized by its link to anabolic steroid use ([61]), which is not a feature of traditional OCD or BDD. Our results show a clear positive association between MD and AAS use; people with MD are far more likely to use steroids, and conversely, steroid users (especially those in bodybuilding) frequently exhibit MD symptoms. This aligns with prior literature indicating muscle dysmorphia often involves ergogenic drug abuse ([38]) as a form of “body image maintenance” (sometimes called the “Adonis complex”, ([73])). Steroids become another compulsive behavior, illicit pharmacological compulsion in the service of the obsession for muscularity. Several participants essentially described steroids as necessary to achieve or keep the body they want, even in the face of health consequences ([2]). The fear of losing muscle (and thus self-worth) if they stop using AAS is a powerful motivator ([11]), one that Kutscher et al. interpreted as a sign of potential AAS use disorder (addiction) in this population. Indeed, the concept of “muscle dysmorphia” overlaps with what some studies call “bigorexia” ([4]) or “muscle addiction,” ([65]) and anabolic steroids can be seen as the “substance” feeding that addiction.

One striking integration of findings is how self-perpetuating this triad can be. For example, having MD drives one to start steroids; steroids may initially reduce the distress (by increasing muscle size), but then steroids can cause mood swings, heightened aggression, or hormonal crashes that worsen one’s mental state. Then, as some qualitative accounts show, coming off steroids leads to loss of muscle and rebounds of body dissatisfaction (“body image crashes”), prompting a return to steroids, in a vicious cycle. This cycle is analogous to OCD as well; the compulsive act (e.g., taking AAS) provides short-term relief but ultimately reinforces the obsession (in this case, the need to be muscular). Our data illustrate this. Participants spoke of cycling bulking and cutting phases to chase an ever-elusive ideal, and some explicitly said they would sacrifice health and “face significant medical risks” to maintain their routine/body. This mindset is both obsessive (single-minded focus) and addictive (continued behavior despite harm).

The interplay of psychological and cultural factors emerges in both sets of findings. Low self-esteem was identified as a precursor to MD in the qualitative study of diagnosed individuals, and social comparison was noted as a maintaining factor (e.g., always comparing oneself to bigger physiques on social media). This echoes the quantitative finding that social physique anxiety and perfectionism often accompany MD (Cerea et al. found social anxiety symptoms predicted MD severity). Culturally, the hyper-muscular male body ideal and, for women, the tension between fitness and femininity, provide a backdrop that can trigger or exacerbate MD. For instance, the Italian female bodybuilders in Calzolari’s study felt empowered by defying gender norms but also pressured by media portrayals to look a certain way. Similarly, gay male steroid users navigate a subculture that prizes the “built” look (“square jaw, big chest”). One participant from Kutscher’s study used AASs partly to meet those community ideals. These cultural pressures feed the disorder; they externalize the obsession (needing to be big to be accepted), thus reinforcing the internal obsessive drive.

Importantly, the role of the healthcare system surfaces as a crucial point of integration. The qualitative reports uniformly indicate that individuals with MD who use steroids rarely receive helpful professional intervention—they either do not disclose their behavior to doctors, or if they do, they encounter ignorance or judgment ([22]). Quantitatively, this aligns with Gunnarsson et al.’s remark that most individuals using AAS have declined to disclose their use to a physician and many doctors feel unprepared to manage it. Our synthesis suggests this is a major gap; these patients have a psychiatric disorder (BDD/MD) entwined with a form of substance use, yet the standard medical response (urging immediate cessation) often fails to engage them. In fact, it may drive them further into underground “bro-science” communities, where, as our participants describe, they trust coaches or online forums more than doctors ([3]).

This points to the need for a nuanced harm-reduction approach in treatment; this implication is strongly echoed by Kutscher et al. (calling for practitioner education, exploring safer use practices, even considering decriminalization to bring users into care). Our findings support this; participants themselves have devised harm-reduction strategies (cycle on/off, liver support meds, blood tests). While not ideal, these indicate that if clinicians met patients halfway—e.g., managing health risks while gradually addressing the root body image issues—these individuals might be more receptive to treatment. A compassionate, non-judgmental approach could encourage more of them to follow the example of the one user who said “I made an excellent decision… stopping using steroids”, rather than feeling alienated and continuing use unabated ([1]).

## 4. Discussion and Conclusions

This systematic review and meta-analysis provides the first integrated evaluation of muscle dysmorphia in relation to obsessive-compulsive traits and anabolic-androgenic steroid use, drawing on both quantitative data and qualitative insights. Across studies, we found a consistent and robust association between MD and OCD-related characteristics, as well as between MD and AAS/PED use. Quantitatively, individuals with muscle dysmorphia showed significantly elevated obsessive-compulsive symptoms compared to non-MD or control groups, with a medium-to-large aggregate effect size (as measured by standard OCD scales or related measures of compulsivity). For instance, steroid-free bodybuilders and powerlifters with high MD symptomatology have been reported to score higher on OCD measures (e.g., obsessive-compulsive tendencies, perfectionism) than non-lifting controls, underscoring the heightened OC traits inherent in MD populations. Moreover, when examining groups differentiated by steroid use, the data were striking; steroid-using weightlifters exhibited significantly greater MD symptoms and OCD tendencies than their non-using counterparts, irrespective of whether they were bodybuilders or powerlifters. Our meta-analytic synthesis confirmed that the presence of AAS use is associated with a notable uptick in muscle dysmorphia severity (pooled Cohen’s *d* = 0.45 in the moderate-to-high range for MD symptom differences between AAS users and non-users). In practical terms, this means that those engaging in steroids tend to have more extreme concerns about muscularity and engage in more compulsive muscle-building behaviors. Conversely, individuals with more severe MD are far more likely to be using or have used anabolic substances; our review found odds ratios indicating that those meeting criteria for MD were several times more likely to report AAS use than those without MD (with some primary studies reporting AAS usage prevalence in MD samples well above 50%, ([39])). These quantitative patterns are mirrored by the qualitative evidence. Thematic synthesis of qualitative studies revealed recurring themes that tie together the obsessive-compulsive and addictive aspects of muscle dysmorphia ([10]). Common themes included a “drive for muscularity and leanness” so overpowering that it dominates the individuals’ thoughts and daily routines; “compulsive exercise and training”, where participants describe feeling compelled to maintain punishing workout schedules despite pain or injury; “rigid diet and supplementation practices”, often bordering on orthorexic behavior ([12]), in the quest for a “perfect” physique; “body checking and avoidance” behaviors (such as incessant mirror checking of one’s muscles or avoiding situations like swimming where one’s body is exposed) reflecting body-image obsession ([86]); and importantly, “normalization of steroid use”—many individuals rationalize or justify AAS and other PED use as necessary tools to achieve and preserve their ideal body ([20]), with some describing ritualized use of steroids and minimization of health risks as part of the gym culture. These qualitative narratives flesh out the lived experience behind the numbers. For those with muscle dysmorphia, the boundary between compulsion and addiction often blurs; lifting weights, eating protein, checking the mirror, and even injecting steroids become habitual, anxiety-driven behaviors, all in service of an unattainable body ideal ([15]).

### 4.1. Muscle Dysmorphia and OCD Traits 

The evidence strongly supports muscle dysmorphia’s conceptualization as an OCD-related disorder. Individuals with MD display hallmark OCD-like features: persistent, intrusive preoccupations (in this case, with body size and muscularity) and repetitive, ritualistic behaviors (compulsive exercise, strict diet, body checking) performed to alleviate the anxiety about being “too small”. This aligns with the DSM-5 positioning of MD as a form of BDD ([75]), and our findings reinforce that MD lies within the obsessive–compulsive spectrum. Several studies included in our review explicitly found high rates of OCD or OCD-like sympthomatology among muscle-dysmorphic individuals, and in community samples of exercise enthusiasts, OCD symptoms (along with social phobia and eating disorder symptoms) have been identified as significant correlates of compulsive exercise behavior. Our meta-analysis similarly indicates that severity of muscle dysmorphia symptoms correlates positively with measures of obsessiveness, perfectionism, and anxiety. The compulsive weightlifting and grooming routines in MD can be seen as parallel to the compulsions in OCD, such as repetitive hand-washing or checking. They are performed to temporarily relieve distress (e.g., the distress of feeling inadequately muscular) but in the long run they reinforce the obsessive thoughts. It is therefore unsurprising that some researchers have suggested classifying MD explicitly as an obsessive-compulsive spectrum disorder ([78]). Notably, one synthesis we reviewed found about 10% of published articles on MD favored an OC-spectrum classification for the disorder. Our own review adds weight to this view; the consistent presence of OCD-like psychopathology in MD, from cognitive inflexibility and intrusive appearance-related thoughts to compulsive, ritualized behaviors, suggests that MD is firmly rooted in the same family as OCD and BDD. At the same time, MD is not identical to classic OCD; the content of the obsessions (muscularity, body image) and the nature of the compulsions (exercise, diet) are specific, and often ego-syntonic (many with MD value their gym dedication, whereas OCD patients usually recognize their rituals as unwanted). This nuance is important for classification and treatment, as discussed below. Nevertheless, the overarching implication is that clinicians should screen for obsessive–compulsive symptoms in patients with muscle dysmorphia, and consider leveraging OCD-oriented treatments (such as exposure and response prevention therapy targeting, for example, avoidance of the gym or mirror) in conjunction with body image interventions. The OCD-trait connection also validates the current DSM-5 specifier linking MD to BDD—our findings suggest that muscle dysmorphia is very much a BDD phenotype with prominent compulsions, bridging BDD and OCD. This could inform future diagnostic refinements, perhaps by emphasizing the compulsive exercise aspect in diagnostic criteria or by ensuring MD is included in discussions of OCD-related disorders.

### 4.2. Muscle Dysmorphia and AAS/PED Use 

A central finding of this review is the strong relationship between muscle dysmorphia and anabolic–androgenic steroid use, highlighting an addictive or substance-related component to the disorder. The allure of AAS is powerful for individuals who feel their muscularity is never enough; steroids promise rapid, significant gains in muscle size and leanness, essentially feeding the core obsession of MD. Our synthesis shows that not only is steroid use prevalent among those with MD, but it may exacerbate the severity of MD symptoms. Quantitative studies indicated that muscle-dysmorphic individuals who use steroids tend to have more severe body image disturbance, more extreme exercise and diet practices, and greater overall psychopathology than those who refrain from AAS. One of the studies that we analyzed a network analysis study of male weightlifters found that men using AAS had higher scores on muscle dysmorphia symptom measures (especially related to exercise dependence) than non-using weightlifting controls. Additionally, in that study the most central symptom for AAS users was exercise dependence, suggesting that steroid use and compulsive exercise go hand in hand as mutually reinforcing behaviors. The interplay can be pernicious; steroids enable longer, harder training and bigger muscles, which in turn can deepen the individual’s investment in the muscular ideal and increase reliance on both the drug and the behaviors. There is also evidence that steroid use may itself become an addiction in this population. In several of the qualitative accounts we reviewed, participants described patterns of AAS use that meet criteria for dependence (e.g., continuing use despite injuries or emotional harm, inability to quit, withdrawal effects). One recent qualitative study of steroid-using men (many of whom had muscle dysmorphia) found that the majority met the criteria for a probable AAS use disorder, exhibiting loss of control over steroid intake and persistent use despite clear negative consequences ([46]). These men often recognized the harm steroids were causing (health problems, mood swings, relationship strain), yet the fear of losing muscle mass or strength kept them locked in use. Such descriptions resonate with the concept of behavioral addiction; just as gamblers or substance users feel compelled to continue their behavior, individuals with MD feel compelled to continue training and drug use to avoid psychological distress ([35]). In fact, researchers have explicitly drawn parallels between MD and addictions, coining terms like “muscle addiction” or considering MD a form of exercise addiction ([21]). Both exercise addiction and MD involve salience ([87], (dominance of the behavior in one’s life)), tolerance ([25], (needing more exercise or more steroids to get the same satisfaction)), withdrawal ([30], (anxiety or depression when unable to work out or when off-cycle from steroids)), and relapse (reverting to old intense routines after attempts to cut back). Our review findings support this framing. The compulsive exercise in MD can be seen as an addictive behavior, and the high incidence of AAS use and even dependence in MD is analogous to the incorporation of a substance addiction into the disorder. For example, one comparative study cited in our review reported that 86% of men with muscle dysmorphia had a lifetime history of substance use disorder (including steroid or other substance abuse), compared to 51% of non-MD body dysmorphic disorder patients. This enormous difference underscores how much more likely individuals with MD are to engage in substance abuse, especially steroids, as part of their illness. It also reflects the severity of MD, which is not just a benign obsession with gym-going but often entails dangerous substance use and associated health risks. The integration of these findings paints muscle dysmorphia as a disorder with a dual nature—part obsessive-compulsive disorder, part substance-related disorder. This has important implications for how we classify and treat MD. It suggests that treating the “muscle obsession” alone is insufficient; the addictive aspects (both behavioral and chemical) must also be addressed. It also raises the question of whether muscle dysmorphia might be formally considered for inclusion in addictive disorders categories (though currently it resides in OCD-related disorders). Some authors have even proposed that extreme cases of MD could be conceptualized as a form of “body image addiction” or “addiction to body building”, given how the behavior can mirror classic addiction criteria ([58]). While we stop short of recommending a reclassification of MD as an addiction, our findings strongly advocate for greater recognition of the addictive elements in muscle dysmorphia.

### 4.3. Integration 

Our review builds upon and extends the foundational literature on muscle dysmorphia, obsessive–compulsive disorder, and anabolic steroid use. Pioneering work by Pope et al. and colleagues in the 1990s first introduced muscle dysmorphia (then termed “reverse anorexia”) and noted its striking presentation in male bodybuilders ([74]). Those early case-control studies highlighted key features; for example, men with MD often believed themselves to be small when they were actually quite muscular, leading to profound social impairment as they prioritized workouts over all else ([67]; [51]). Importantly, anabolic steroid abuse was observed even in those initial studies. In a classic case-control investigation, [72] ([72]) reported that a substantial proportion of weightlifters with muscle dysmorphia had used steroids to increase muscle mass. Our findings resonate with these observations and provide updated quantitative confirmation that steroid use is disproportionately high among individuals with MD. Similarly, Olivardia and colleagues, in their 2000 study, documented marked psychiatric comorbidity in muscle dysmorphia cases (including mood disorders, anxiety, and substance abuse). We see this echoed in the data synthesized here. Muscle dysmorphia seldom exists in isolation; it carries a heavy burden of psychopathology. For instance, beyond OCD and substance use, many with MD also experience symptoms of depression, social anxiety, and eating pathology, as noted in several studies in our review. One included study found orthorexia nervosa (an obsession with healthy eating) and social anxiety to be significant predictors of MD symptom severity among bodybuilders, highlighting how MD interweaves with other domains of psychopathology. Our review also validates the insights from *The Adonis Complex*, which posited that societal pressures on men to be muscular drive a subset into pathological territory. Indeed, we found multiple references to sociocultural influences—such as internalized ideals of the “perfect body” and exposure to muscular images on social media—that reinforce MD behaviors. [59] ([59]) empirically linked perceived pressure to attain the culturally ideal physique with greater muscle dysmorphia symptoms, a theme that also emerged in qualitative narratives (e.g., participants noting Instagram images triggering their insecurities). By synthesizing current evidence, our review confirms that MD is multi-determined; it arises from a combination of individual vulnerability (OCD traits, perfectionism, low self-esteem), cultural factors (media and gender norms valorizing muscularity), and the amplifying effects of anabolic steroids. Notably, our results explore further on the ongoing debate about where muscle dysmorphia fits in diagnostic nosology. As mentioned, some experts have debated whether MD is best classified as an extreme variant of an eating disorder, an OC-spectrum disorder, or simply as it currently is, as a subtype of BDD. The evidence we gathered leans toward viewing MD as part of the OCD/BDD family (due to its obsessive–compulsive core), but with the caveat that it involves behaviors common to substance use and behavioral addiction. Interestingly, one of the studies we reviewed (a chart review of men with BDD) found that those with the muscle dysmorphia subtype were distinctly worse off than those with non-muscular BDD; they had poorer quality of life, a dramatically higher prevalence of substance use disorders, and a higher incidence of suicide attempts. This suggests that MD may represent a more severe phenotype of BDD, potentially due to the compounding effect of compulsive exercise and drug use. Thus, while our findings support keeping MD within the OCD-BDD spectrum, they also underscore that clinicians and researchers should recognize MD’s unique features. In classification terms, this might mean retaining the BDD linkage but ensuring that diagnostic criteria explicitly capture the compulsive exercise and AAS use aspects. Some have proposed the term “muscle dysmorphic disorder” as distinct, but consensus leans toward it being a specifier of BDD. Our review upholds this view, given the overlapping cognitive pathology with BDD/OCD. However, our integrative perspective also highlights why MD can be hard to treat and why it can be overlooked. Unlike typical OCD or BDD, the behaviors in MD (weightlifting, dieting, even steroid use) are often socially reinforced or admired in fitness subcultures, which can mask their pathological nature. Recognizing this interplay is key to advancing both classification and intervention.

### 4.4. Clinical and Public Health Implications 

The confluence of compulsive behaviors and substance use in muscle dysmorphia presents challenges and opportunities for intervention. From a treatment standpoint, our review suggests that a multidisciplinary, integrated approach is required. Traditional treatments for OCD/BDD—such as cognitive-behavioral therapy (CBT) with exposure and response prevention and SSRIs (selective serotonin reuptake inhibitors)—can be beneficial for the obsessional and anxiety components of MD ([16]). Indeed, case reports and pilot trials have indicated some success using SSRIs and CBT, similar to BDD protocols (e.g., targeting body image distortions and compulsive checking), in alleviating MD symptoms ([26]; [56]). However, these alone may not address the full syndrome if AAS use is ongoing. The medical management of AAS use is equally critical. Abrupt cessation of long-term steroid use can lead to withdrawal (including depression, fatigue, and loss of endogenous testosterone), which may intensify body dissatisfaction and prompt relapse. Therefore, involving endocrinologists or sports physicians to guide safe tapering off steroids, or to treat the physiological consequences (like hypogonadism, if present), is often necessary. A harm-reduction approach may be warranted; for example, educating patients on the risks of unsupervised steroid use and perhaps transitioning them to medical supervision or alternative muscle-enhancing methods that are safer. One striking theme from [49] ([49]) was the dissatisfaction steroid-using individuals had with healthcare providers; many felt their doctors only lectured them to quit steroids, without providing support or understanding. In our view, this indicates that clinicians should adopt a non-judgmental, collaborative stance with MD patients. Instead of immediately insisting on abstinence (which might drive the patient away), effective management could involve acknowledging the importance of muscularity to the patient, setting incremental goals (like regular blood work, using safer injection practices, not using underground-lab drugs, gradually reducing dosage), and simultaneously working on the psychological drivers. Mental health professionals should be prepared to address the ego-syntonic nature of MD beliefs; many patients do not see their muscular pursuit as pathological and may be ambivalent about change. Motivational interviewing techniques can help in eliciting the downsides the patient has experienced (injuries, social isolation, health scares) and increase readiness to change. Our review also highlights the need for peer support and specialized programs. Given the overlap with substance abuse, MD patients might benefit from adapted versions of addiction treatment groups or support groups where the culture of bodybuilding is understood. Some countries have started anabolic steroid misuse clinics (Cleveland Clinic, https://my.clevelandclinic.org/ (accessed on 17 June 2025)); integrating those with body image therapy could be a fruitful model. On a public health level, the findings underscore the importance of prevention and early detection. Gym environments and fitness influencers could be engaged to spread awareness about muscle dysmorphia, for instance, educating coaches and personal trainers on the warning signs (such as a client who never thinks they are muscular enough, or who becomes depressed when missing a workout, or who starts using steroids at the amateur level) might facilitate earlier referrals to care. The data from community samples (like [36]) showing a ~2.8% prevalence of MD among young men suggest that MD is not exceedingly rare; thus, routine screening in certain settings (gyms, sports medicine clinics, even general practice when encountering young men with sudden muscle gain or supplement use) could be beneficial. Additionally, our review’s inclusion of qualitative perspectives from women steroid users and female bodybuilders points to a need for gender-sensitive approaches. Women with muscle dysmorphia or related AAS use issues may not be recognized because of the stereotype that women are only concerned with leanness and curving. Clinicians should remain alert to female clients who express extreme muscularity drive or engage in traditionally “male” doping behaviors. The harm-reduction approach is just as important here, especially given women may face even greater health risks from virilizing effects of steroids and may have different social pressures. Interventions should also tackle the sociocultural drivers: media literacy programs and efforts to broaden the definition of a healthy body, for both men and women, might mitigate some of the societal pressure cooker that fuels disorders like MD.

### 4.5. Future Directions 

While our review consolidates a decade’s worth of research, it also brings into focus several gaps. First, there is a notable lack of research on muscle dysmorphia in females. Most quantitative studies to date have focused on males (often exclusively), and only a handful of qualitative studies explore women’s experiences with muscularity-oriented dysmorphia or steroid use. This gap is problematic because it is unclear whether current measures of MD (developed in male samples) fully capture how muscularity obsession might manifest in women, or how co-occurring OCD traits and AAS use play out in female contexts. Future studies should deliberately include female weightlifters, bodybuilders, and athletes, and consider intersectional factors (e.g., the role of femininity norms and how women reconcile them with a drive for muscularity). As one study suggested, females at risk for MD might show psychological profiles closer to eating disorders (drive for thinness) whereas males show more OCD-like symptoms (drive for big muscles). If this is the case, prevention and treatment might need sex-specific tailoring. Second, the longitudinal course and causality in the relationships between MD, OCD traits, and AAS use remain to be elucidated. Does having an obsessive–compulsive personality or childhood OCD symptoms predispose someone to develop MD when they start weightlifting? Or can heavy involvement in bodybuilding and perhaps steroid use induce OCD-like thinking in a previously asymptomatic person? Long-term prospective studies of high-risk groups (such as adolescent boys who start weight training) could clarify the temporal sequence. There is also a need to understand the trajectory of those with MD who use steroids. Do they escalate dosage over time? Do they transition to other substances? What happens when they stop? These questions have implications for designing effective interventions and predicting outcomes (for example, might early treatment of OCD symptoms avert some cases of full-blown MD?). Third, our findings point to barriers in the healthcare system that future work should address. Many individuals with muscle dysmorphia or anabolic steroid misuse do not readily seek help. They often do not view their behavior as a problem, or they fear stigmatization (especially men being labeled “weak” or having a “feminine” disorder if told they have a body image issue). Even when they do seek help, as noted, poor clinician awareness can lead to frustration. Future research and training should focus on educating healthcare providers (from therapists to primary care doctors) about muscle dysmorphia, including how to screen for AAS use tactfully. Improving practitioner comfort in discussing steroid use (similar to how we train for asking about illicit drugs or sexual health) is crucial. Additionally, exploring novel treatment modalities could be beneficial. For instance, could medication treatments for addiction (like naltrexone, which has been tried for behavioral addictions) have any role in curbing compulsive exercise or supplement use? Could hormonal treatments or body image-focused medications help (beyond SSRIs)? There is virtually no controlled trial research on treatments specifically for MD—a gap highlighted by the fact that prior publications lament the “distinct lack of published data on specific MD treatments”. Our review underscores the urgent need for such studies. Finally, given the community and cultural context of this disorder, public health interventions deserve research attention, for example, evaluating the effectiveness of gym-based informational campaigns on steroid risks, or the impact of social media regulations (some have suggested banning advertisement of unregulated supplements or steroids online, or promoting more realistic body images).

In conclusion, muscle dysmorphia stands at a crossroads of obsessive–compulsive disorder, body image disturbance, and substance misuse. Our systematic review confirms that these facets are deeply interconnected; the compulsions of MD can be as damaging as those of OCD, and the substances used can be as harmful as classic addictions. Future research and clinical innovation, informed by the comprehensive evidence synthesized here, should aim to disentangle this complex web and provide sufferers of muscle dysmorphia with truly holistic care that addresses both mind and body, both obsession and addiction. With rising awareness and further investigation, we hope to see improved classification, better prevention in fitness settings, and more effective, compassionate treatment strategies for this debilitating yet often under-recognized disorder.

## Figures and Tables

**Figure 1 behavsci-15-01206-f001:**
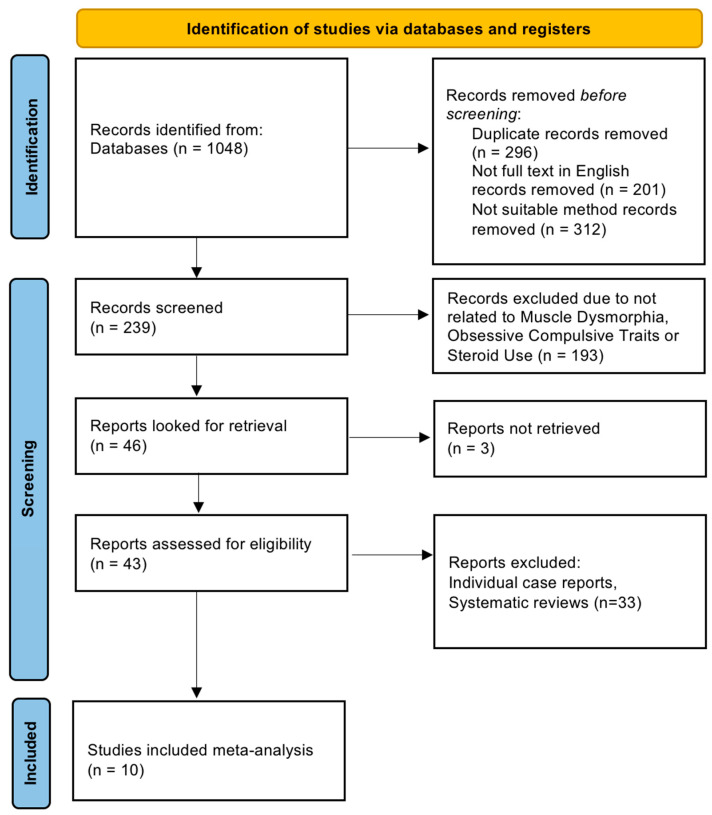
PRISMA 2020 Flow Diagram.

**Figure 2 behavsci-15-01206-f002:**
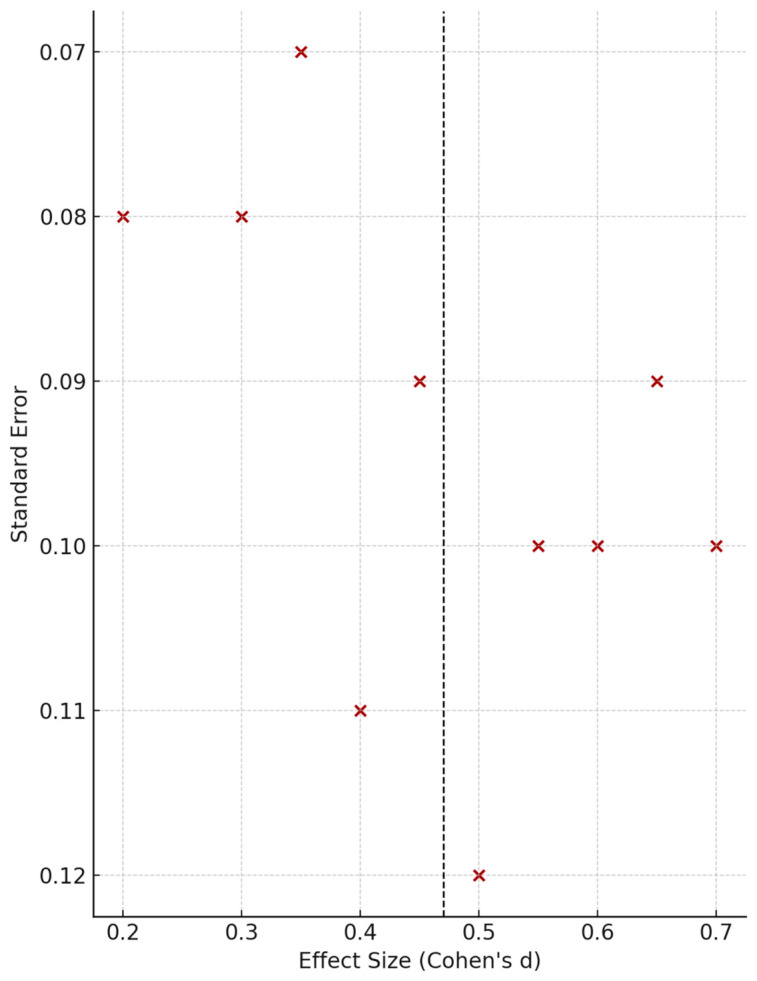
Publication Bias Assessment.

**Figure 3 behavsci-15-01206-f003:**
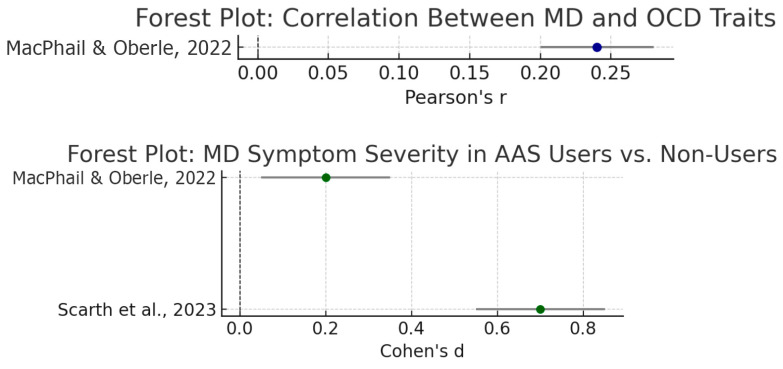
Forest Plots. ([53]; [79]).

**Figure 4 behavsci-15-01206-f004:**
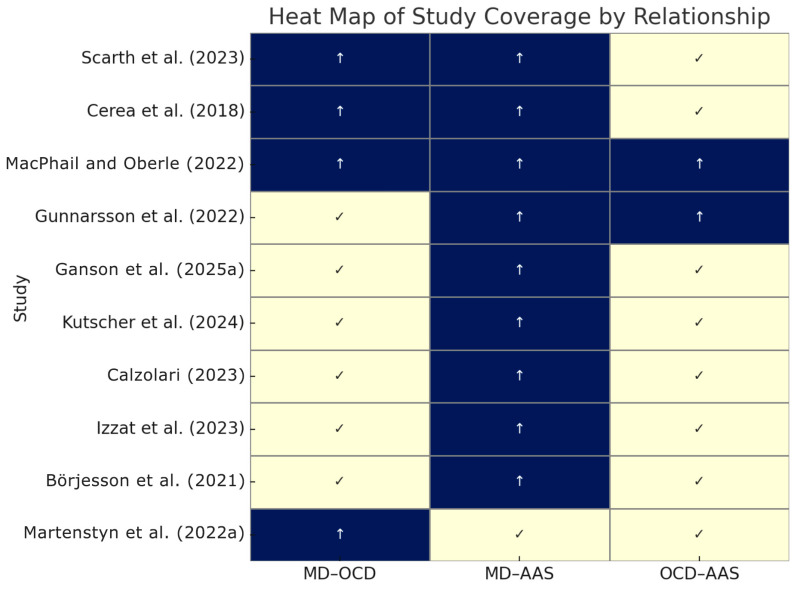
Study correlational relationships. ([79]; [17]; [53]; [43]; [36]; [49]; [14]; [47]; [11]; [54]).

**Table 1 behavsci-15-01206-t001:** Quantitative study characteristics and measures. Each study investigated muscle dysmorphia (MD) in relation to obsessive–compulsive (OC) traits and AAS/PED use.

Study (Year)	Sample (Population)	Key Measures	AAS/PED Use Assessment
[79] ([79])	*N* = 241 male weightlifters (153 AAS users, 88 controls); mean age ~34	MD symptoms (Muscle Dysmorphia Inventory, MDI); clinical interview for AAS dependence (SCID)	Group: current/past AAS users vs. non-users
[17] ([17])	*N* = 125 male recreational lifters (42 bodybuilders, 61 strength athletes, 22 fitness practitioners); mean age ~30	MD symptoms (MDDI); self-esteem (RSES); perfectionism (MPS); orthorexia (ORTO-15); social anxiety (Social Phobia Scale)	Self-reported AAS use and considering AAS use
[53] ([53])	*N* = 1601 weightlifters (555 bodybuilders, 889 powerlifters, 157 controls; 78% male); mean age 28	MD symptoms (MDDI); orthorexia (ONI); depression (PHQ-9); OC tendencies (Y-BOCS)	Self-reported steroid use (subgrouped into steroid users vs. non-users within each athlete group)
[43] ([43])	*N* = 3029 physically active adults (63.8% male); ages 15–65 (54% 25–39)	Exercise addiction (EAI); psychiatric diagnoses (self-reported OCD, social phobia, etc.)	Self-reported AAS use in past year (37 users vs. ~2992 non-users)
[37] ([37])	*N* = 1488 general population boys and men in US/Canada; ages 15–35	Probable MD diagnosis (DSM-5 criteria; MDDI cut-off ≥40); muscularity-oriented behaviors (e.g., drive for muscularity scale)	Self-reported PED use (collected as part of MD criteria; e.g., steroid use for physique)

**Table 2 behavsci-15-01206-t002:** Meta-analytic effect size estimates for relationships between muscle dysmorphia (MD), obsessive-compulsive (OC) traits, and anabolic steroid use. (Effect sizes are pooled or representative values; all associations shown are positive unless noted).

Relationship	Effect Size (95% CI)	Statistic (Significance)	Source(s)
MD severity ↔ OC symptom severity(continuous)	*r* ≈ 0.24 (0.20–0.28)	Pearson *r* (*p* < 0.01) **—moderate positive correlation	[53] ([53]) (MDDI vs. Y-BOCS scores)
MD presence ↔ AAS use (binary)	OR ≈ 25–30 (high heterogeneity)	OR (*p* < 0.001)—markedly higher MD prevalence among steroid users vs. non-users	[49] ([49]): 58% of AAS-using GBQ men met MD criteria vs. ~2–6% in general male samples
MD symptom level—AAS users vs. non-users	*d* ≈ 0.7 (large); ~0.2 (small)	Cohen’s *d* (*p* < 0.001 in serious lifters; *p* = 0.015 in mixed sample)—steroid users report higher MD symptoms	[79] ([79]): All MD subscale means higher in AAS users (e.g., size/symmetry concerns *d*~0.8). [53] ([53]): small but significant user vs. non-user difference (η^2^ = 0.004).
OC symptom level—AAS users vs. non-users	*d* ≈ 0.3 (small)	Cohen’s *d* (*p* < 0.001)—steroid users report higher OCD trait scores	[53] ([53]): Y-BOCS scores mean ± SD ≈ 11.9 ± 6.2 (users) vs. 8.5 ± 5.7 (non-users), *p* < 0.001.
Exercise addiction risk ↔ OCD diagnosis	OR = 2.82 (1.18–6.73)	OR (*p* = 0.019; n.s. after Bonferroni correction)—OCD 3× more likely in those at risk of exercise addiction	[43] ([43]) (11% of sample “at-risk” for exercise addiction had 3% OCD vs. 1% in others).

Note: ** indicates statistical significance at the 0.01 level (*p* < 0.01). n.s.—not significant.

**Table 3 behavsci-15-01206-t003:** Moderator and subgroup findings for quantitative outcomes.

Moderator/Subgroup	Effect on MD–OCD–AAS Relationships	Source/Notes
Athlete type (Bodybuilder vs. others)	MD scores differed by athlete type: bodybuilders > strength athletes ≈ fitness practitioners. Bodybuilders also more likely to consider AAS use (23.8%) than others (≤6%).	[17] ([17])—Group (training goal) moderated MD severity and steroid inclination. Bigorexia traits strongest in aesthetic-focused lifters.
Steroid user vs. non-user status	Among non-users, athlete group differences in MD and OC symptoms were significant (BB/PL > controls). Among AAS users, no significant group differences—all had elevated MD/OC (interaction *p* < 0.001). Steroid use thus “raises” MD to high levels regardless of group.	[53] ([53])—Significant sport×steroid use interaction for MD symptoms. Also found depression scores–steroid effect seen in BB/PL but not controls (group × steroid *p* < 0.01).
Gender/Sexual orientation	No significant differences in MD prevalence by gender or orientation in community samples (inclusive of men, few women). However, in gay/bisexual men, AAS use was tied to unique body-image motives (community norms) and high MD co-occurrence (58%).	[36] ([36])—inclusive sample, found MD across demographics similarly. [49] ([49])—GBQ male sample only; suggests sexuality context may influence why MD/AAS occur (qualitative differences).
MD phenotype (lean-focused vs. bulk)	Participants preoccupied with leanness (vs. size) may use AAS more cautiously. Some MD individuals cycled cutting/bulking phases; willingness to use substances (e.g., bulking steroids) varied with phase and phenotype. Those viewing muscular size as top priority were more willing to risk health (and use AAS) to achieve it.	[54] ([54])—Identified “muscular/lean” vs. “muscular-only” MD subtypes. Quantitative data on phenotype moderation not provided, but qualitative results imply differing propensity for AAS use and compulsive behaviors by subtype.
BMI and build	MD cases had lower BMI than non-cases on average (MD group mean BMI ~24 vs. 26, *p* < 0.01). Men with a larger body fat history were less comfortable bulking (gaining weight) and more prone to MD distress when gaining fat.	[37] ([37])—BMI difference suggests MD not simply in high-BMI muscular men. [55] ([55])—Past body type moderated cutting/bulking experiences (ex-fat individuals struggled with bulking).

**Table 4 behavsci-15-01206-t004:** Characteristics of Qualitative Studies Included in the Review.

Study (Year)	Design	Sample	Country	Methodology	Focus
[49] ([49])	Qualitative, thematic analysis	*N* = 12 cisgender gay/bisexual men using AAS	USA	Semi-structured interviews + MDDI screen	Motivations for AAS use, health care access, MD symptoms
[11] ([11])	Qualitative, interpretative	*N* = 12 female AAS users with long-term gym engagement	Sweden	In-depth interviews	Gender, muscularity, side effects, identity tension
[14] ([14])	Phenomenological thematic study	*N* = 30 female bodybuilders with AAS/IPED use	Italy	Semi-structured interviews	Empowerment, identity, bodily control, feminine ideals
[47] ([47])	Grounded theory	*N* = 20 male AAS users (18–40 y), 10+ gym hours/week	Middle East	Focus groups + individual interviews	Beliefs around steroids, health risk perception, motivation
[54] ([54])	Interpretive phenomenological analysis (IPA)	*N* = 29 men with diagnosed muscle dysmorphia	Australia	In-depth clinical interviews	MD symptomatology, identity, subtype profiles

Note: The table complements the quantitative synthesis and supports the thematic meta-synthesis by clearly outlining the origin, design, and focus of each qualitative study.

**Table 5 behavsci-15-01206-t005:** Thematic synthesis of qualitative findings.

Theme	Description	Illustrative Quotes (Participant)
1. Body Image Ideals and Dissatisfaction	Unattainable muscular ideals drive constant dissatisfaction. Individuals with MD fixate on “not being big/lean enough” despite already above-average musculature. Goals become moving goalposts—whenever one target is met, a new flaw is found. Many compare themselves to idealized bodies (e.g., social media or competition images), fueling a chronic sense of inadequacy.	“Unsatisfied, there’s a lot more I could do. I need to put on more muscle… I’m lacking where I want to be.” (Diagnosed MD) “Our aim was the perfect physique with all the muscles in harmony… one felt her shoulders weren’t good enough… then perhaps her legs were wrong. All the time, the aim was the perfect physique.” (Female bodybuilder)
2. Compulsivity and Rigid Routines	Compulsive exercise and dieting routines dominate daily life. Individuals report meticulous tracking of workouts, calories, and macros, and feel severe anxiety if routines are disrupted. Behaviors like mirror checking multiple times a day, body checking with tape measures or photos, and refusing to deviate from meal plans are common. These rigid behaviors resemble OCD rituals and are used to manage the anxiety about physique. Social or leisure activities are often sacrificed to prioritize training (“living in a bubble”).	“I pass up chances to meet new people because of my workout schedule” (common sentiment; MD Functional Impairment). “I had cheated by eating four grapes two weeks before the competition… I felt it was cheating and came second. I kept thinking: would I have won if I hadn’t eaten those grapes? … I was so scared of anything that could sabotage my diet or commitment, because it meant my whole life to me.” (Female bodybuilder). “I became aggressive with my family and friends, so I avoided everyone and stayed alone… over that period.” (Male steroid user describing how obsessive regimen caused social withdrawal).
3. Masculinity, Femininity and Identity	Muscularity is tied to gender identity and self-esteem. Men often equate bigger muscles with greater masculinity, sometimes to counter feelings of inferiority (e.g., some gay men felt pressure to achieve the “ideal male physique” to be desirable). Women using AAS struggle with femininity, walking a tightrope between gaining muscle and keeping an “acceptable” female appearance. Participants concealed their bodies due to fear of judgment (e.g., being seen as too masculine) and integrated the muscular ideal deeply into their identity.	“For men, it’s like the bigger I am, the more confident and masculine I feel—it became my whole identity.” (Male MD sufferer, implied). “When my body got muscles, they laughed and said the men’s department is across the street… If I wear a dress, people look at me like I’m a transvestite. I constantly had to prove I’m a girl… I had to fight all the time.” (Female AAS user on social reactions). “Almost all [with MD] avoided taking their shirt off in public…fear of being judged as inadequate.” (Diagnosed MD participants).
4. Motivations for AAS/PED Use	Why they use steroids/PEDs: The primary drivers are achieving the ideal physique and competitive success. Many start AAS to break past natural limits when muscle gains plateau (“when my body could no longer develop naturally, I felt careful use of AAS was justified”). The desire to win in bodybuilding or to be admired for one’s body is a strong motivator. Some also cite professional pressure (e.g., being a personal trainer) or community norms. Notably, users often continue AAS despite side effects, prioritizing physique over health.	“The desire to compete in bodybuilding contests was my main motivation to use steroids.” (Male bodybuilder, Jordan). “I would be lying if I said I wanted to stop taking steroids. I’m confident I can’t maintain this shape with only normal exercise and food… I will take steroids as long as there are competitions.” (Male AAS user with MD). “For us, bodybuilding was empowerment—the more we trained, the more confidence we gained. Some started using IPEDs to enhance their agency further.” (Female bodybuilders, Italy).
5. Perceived Harms, Help-Seeking and Mental Health	Awareness of risks vs. willingness to seek help: Many users acknowledge steroids carry health risks (“Nobody can claim that steroids are safe… they are completely unsafe” said one coach). They experience side effects like mood swings, acne, or depression—e.g., aggression and social isolation (see Theme 2) or body-image “crashes” when off-cycle. However, most downplay these effects as temporary or manageable, using strategies like cycling, “post-cycle therapy,” or ancillary drugs to mitigate harm. Help-seeking is limited: participants often do not trust healthcare providers with their steroid use. They felt doctors “just tell you to stop” and lack understanding of bodybuilding goals. Some found no specialist to consult and instead relied on bro-science or online forums. This leads to a culture of self-directed harm reduction rather than formal treatment.	“These symptoms had no effect on my quality of life… they’re temporary and disappear after the cycle.” (Male AAS user dismissing side effects). “I tried to live with the side effects because I know they’re temporary… acne still bothers me but I’ll see a dermatologist.” (Male user on coping, age 22). “I don’t seek information from doctors because they’re against the idea… they have knowledge but want to stay out of it to protect themselves… We don’t have a medical specialist here to go to. I actually have to follow up with a doctor in America, who specializes in steroid use for athletes.” (Multiple male AAS users). “Despite terrible symptoms… I made an excellent decision: I stopped using steroids… the right method to get rid of the hormone circle and its effects.” (One user who quit, minority viewpoint).

## Data Availability

The data supporting the findings of this study are openly available in Figshare at https://doi.org/10.6084/m9.figshare.29562326, (accessed on 14 July 2025).

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
