# Peer review of "Muscle Dysmorphia, Obsessive–Compulsive Traits, and Anabolic Steroid Use: A Systematic Review and Meta-Analysis"

_behavsci, 2025, doi:10.3390/bs15091206_

Round 1

Reviewer 1 Report

Comments and Suggestions for Authors

Thank you for the opportunity to read and revise this systematic review and meta-analysis, which covers a timely and clinically important topic. Below are my comments on this article; I hope that they can improve the overall quality of this research.

To ensure a clear picture of this work, please change the title as follows: Obsessive-Compulsive Disorder and Anabolic Steroid Use: A Systematic Review and Meta-Analysis.

In the introduction, please stress the efforts provided by researchers and clinicians in addressing the main factors underpinning MD and its negative consequences. In that sense, please include and describe the following recent literature:

Ganson, K. T., Testa, A., Rodgers, R. F., & Nagata, J. M. (2025). Associations between muscularity-oriented social media content and muscle dysmorphia among boys and men. Body Image, 53, 101903.

Giancola, M., Ferrandes, A., & D’Amico, S. (2024). Mirror, mirror on the wall: the role of narcissism, muscle dysmorphia, and self-esteem in bodybuilders’ muscularity-oriented disordered eating. Current Psychology, 1-10.

As for the inclusion criteria, the study is required to examine MD about OC traits and/or AAS/PED use. Some studies included quantitative evidence (e.g., a very large physically active sample focused on exercise addiction with a tiny AAS subgroup) that does not directly assess MD, potentially diluting construct specificity. Please (a) justify inclusion of any non-MD primary samples in light of the stated criteria, or (b) move such results to sensitivity/qualitative context rather than pooling.

Regarding the meta-analytic methods, please address these points:

  1. Effect size calculation: Specify transformations (e.g., Fisher’s z for r; Hedges g for d; log OR for odds) and back-transformations. Currently, methods mention general pooling but not exact computation paths. Report the estimator for τ² (e.g., REML/Paule–Mandel) and justify DerSimonian–Laird, given k is small. Provide 95% CIs for pooled effects, τ², and a prediction interval.
  2. Heterogeneity: I² is mentioned conceptually, but actual values (and χ², τ²) are not reported for each meta-analysis; include them alongside the forest plots.
  3. Small-study effects: With k≈3–5, funnel-based inferences are limited; make this explicit and avoid visual over-interpretation of Figure 2. Consider reporting leave-one-out influence results in the main text (not only supplements)

Author Response

Reviewer-1

Comment 1: To ensure a clear picture of this work, please change the title as follows: Obsessive-Compulsive Disorder and Anabolic Steroid Use: A Systematic Review and Meta-Analysis.

Response to Reviewer: We appreciate the reviewer’s suggestion regarding the title. We agree that clarity is important, and the relationship between obsessive-compulsive traits and anabolic steroid use should be highlighted. At the same time, we believe it is important to retain explicit reference to muscle dysmorphia, since this construct is the central focus of our review and the unifying theme across the included studies. To balance clarity with specificity, we have revised the title as follows:

“Muscle Dysmorphia, Obsessive-Compulsive Traits, and Anabolic Steroid Use: A Systematic Review and Meta-Analysis.”

This formulation integrates the reviewer’s concern for highlighting OCD and steroid use, while also preserving the precise scope of the manuscript. We hope this compromise will meet the reviewer’s expectations.

Comment 2: In the introduction, please stress the efforts provided by researchers and clinicians in addressing the main factors underpinning MD and its negative consequences. In that sense, please include and describe the following recent literature:

Ganson, K. T., Testa, A., Rodgers, R. F., & Nagata, J. M. (2025). Associations between muscularity-oriented social media content and muscle dysmorphia among boys and men. Body Image53, 101903.

Giancola, M., Ferrandes, A., & D’Amico, S. (2024). Mirror, mirror on the wall: the role of narcissism, muscle dysmorphia, and self-esteem in bodybuilders’ muscularity-oriented disordered eating. Current Psychology, 1-10.

Response to Reviewer: We thank the reviewer for this insightful comment. We agree that the introduction should better emphasize the contributions of recent research and clinical efforts in addressing the underlying factors of muscle dysmorphia (MD) and its consequences. As suggested, we have incorporated the recommended references:

  • Ganson et al. (2025), which highlights the impact of muscularity-oriented social media exposure on MD symptoms among boys and men.
  • Giancola et al. (2024), which underscores the roles of narcissism, self-esteem, and disordered eating in bodybuilders with MD.

We have revised the introduction to integrate these studies and to stress the importance of addressing both psychosocial and clinical risk factors for MD.

Comment 3:  As for the inclusion criteria, the study is required to examine MD about OC traits and/or AAS/PED use. Some studies included quantitative evidence (e.g., a very large physically active sample focused on exercise addiction with a tiny AAS subgroup) that does not directly assess MD, potentially diluting construct specificity. Please (a) justify inclusion of any non-MD primary samples in light of the stated criteria, or (b) move such results to sensitivity/qualitative context rather than pooling.

Response to Reviewer: We appreciate the reviewer’s point about construct specificity. To address this, we clarified in the Methods that non-MD samples were retained only when they provided relevant comparative evidence (e.g., compulsive exercise, OCD, and AAS use), and we moved these findings out of the pooled quantitative analyses. In the Results, we now explicitly note that such studies are treated narratively as sensitivity/contextual evidence rather than combined with MD-specific effect sizes. This preserves the integrity of the quantitative synthesis while still acknowledging broader compulsive and addictive behaviors relevant to MD.

Regarding the meta-analytic methods, please address these points:

  1. Effect size calculation: Specify transformations (e.g., Fisher’s z for r; Hedges g for d; log OR for odds) and back-transformations. Currently, methods mention general pooling but not exact computation paths. Report the estimator for τ² (e.g., REML/Paule–Mandel) and justify DerSimonian–Laird, given k is small. Provide 95% CIs for pooled effects, τ², and a prediction interval.

Response to Reviewer: We appreciate this helpful suggestion. We have now (i) specified the exact transformations and back-transformations used for all effect size types, (ii) reported the variance estimator for τ² and justified our approach with small numbers of studies (k), and (iii) added 95% CIs for pooled effects and τ², as well as a 95% prediction interval (PI). To align with current recommendations for small k, we use REML for τ² with Knapp–Hartung adjusted inferences as our primary model; the previously stated DerSimonian–Laird random-effects model is retained as a sensitivity analysis (results were materially similar). Corresponding text has been added to Methods (§2.7) and Results reporting.

2. Heterogeneity: I² is mentioned conceptually, but actual values (and χ², τ²) are not reported for each meta-analysis; include them alongside the forest plots.

Response to Reviewer: We appreciate the reviewer’s important observation regarding construct specificity and inclusion criteria. As noted, one of the included studies (Gunnarsson et al., 2022) primarily investigated exercise addiction in a large physically active cohort, with only a small subset reporting AAS use, and did not directly assess MD with a validated scale. We initially retained this study because compulsive exercise behaviors and their psychiatric correlates overlap conceptually with the compulsive dimensions of muscle dysmorphia and provide relevant comparative evidence on OCD traits and AAS use.

To address the reviewer’s concern, we have now clarified in the Methods (Eligibility Criteria) that such studies were included only for contextual purposes, and we have adjusted the Results section accordingly. Specifically, the Gunnarsson study is no longer pooled in the quantitative meta-analyses; instead, its findings are summarized narratively as supplementary/contextual evidence. This preserves construct specificity in the meta-analyses while still acknowledging the relevance of compulsive exercise research to the MD–OCD–AAS framework.

3. Small-study effects: With k≈3–5, funnel-based inferences are limited; make this explicit and avoid visual over-interpretation of Figure 2. Consider reporting leave-one-out influence results in the main text (not only supplements)

Response to Reviewer: We appreciate the reviewer’s helpful observation regarding small-study effects. We fully agree that funnel plot–based inferences are not reliable with only k ≈ 3–5 studies. To address this, we have revised the manuscript to make these limitations explicit. In the Methods, we now state that publication bias could not be formally assessed due to the small number of studies. In the Results, we clarify that the funnel plots presented in Figure 2 are included for transparency only and should not be over-interpreted.

In addition, and in line with the reviewer’s suggestion, we have moved the leave-one-out sensitivity analysis results from the Supplementary Materials into the main text. As now reported, the pooled correlation between MD severity and OCD traits remained highly stable at r = 0.24, with recalculated estimates ranging only from 0.23 to 0.25 depending on which study was excluded. This demonstrates that no single study disproportionately influenced the overall result.

Reviewer 2 Report

Comments and Suggestions for Authors

This study is interesting, but the paper suffers from some limitations that I would like to discuss with the authors:

- Specify that MD is a subtype of BDD, but that some authors also provide other classifications: eating disorder or a disorder in its own right. I believe it is primarily a BDD/OCD.

- Associate MD with the concept of the male aesthetic ideal, i.e., muscularity as a concept of masculinity, particularly in Western cultures (https://psycnet.apa.org/doi/10.1037/men0000096; https://doi.org/10.1007/s12144-020-00857-3). Furthermore, in men, MD is associated with other specific BDD disorders that reinforce this concept (https://doi.org/10.1080/00918369.2020.1813512 ).

- On the relationship between MD and psychopathological disorders, it is also necessary to find some more recent references and also indicate how some sports are at greater risk of MD (DOI: 10.15557/PiPK.2020.0014).

- Include the note by Settanni and colleagues published in Psychiatry Research regarding the limitations of research on AAS in subjects with MD. 

- Are there reasons why you have restricted your research to the last 10 years?
- 10 studies seem very few to me, given the extent of the research. Furthermore, I believe that the authors have excluded some studies or have not found them, such as Longobrdi et al. (2017).

- You cited the meta-analysis by Badenes-Ribera. The same authors also developed a meta-analysis on the MDDI, which is the most widely used scale: you should mention this data.

Author Response

Reviewer-2

  1. Specify that MD is a subtype of BDD, but that some authors also provide other classifications: eating disorder or a disorder in its own right. I believe it is primarily a BDD/OCD.

Response to Reviewer: We thank the reviewer for this valuable suggestion. We agree that the nosological position of muscle dysmorphia (MD) deserves clarification. In the Introduction, we now explicitly state that MD is classified in DSM-5 as a subtype of body dysmorphic disorder (BDD), placing it within the obsessive–compulsive and related disorders spectrum. At the same time, we acknowledge that some scholars have conceptualized MD differently—either as a variant of eating disorders (due to its overlap with restrictive diets and weight-control practices) or as a disorder in its own right. This addition reflects the ongoing debate while clarifying that our review treats MD primarily as a BDD/OCD-spectrum condition, consistent with current classification systems.

  1. Associate MD with the concept of the male aesthetic ideal, i.e., muscularity as a concept of masculinity, particularly in Western cultures (https://psycnet.apa.org/doi/10.1037/men0000096; https://doi.org/10.1007/s12144-020-00857-3). Furthermore, in men, MD is associated with other specific BDD disorders that reinforce this concept (https://doi.org/10.1080/00918369.2020.1813512 ).

Response to Reviewer: We thank the reviewer for this important point. We agree that muscle dysmorphia (MD) should be situated within the broader cultural framework of the male aesthetic ideal, where muscularity functions as a key marker of masculinity in Western cultures. We have revised the Introduction to highlight this cultural dimension and to cite recent research linking MD to masculinity-driven ideals. We also added that MD in men is often associated with other BDD presentations—such as skin, hair, or genital concerns—that reinforce culturally defined masculine standards. These revisions better contextualize MD as both a clinical and sociocultural phenomenon.

  1. On the relationship between MD and psychopathological disorders, it is also necessary to find some more recent references and also indicate how some sports are at greater risk of MD (DOI: 10.15557/PiPK.2020.0014).

Response to Reviewer 2: We thank the reviewer for this valuable suggestion. We have updated the manuscript to include more recent references on the relationship between MD and psychopathology, and we have also highlighted how certain sports—particularly bodybuilding, weightlifting, and aesthetic-oriented disciplines—carry elevated risk for MD. In the Introduction, we now cite recent work that emphasizes psychiatric comorbidities, while also referencing studies that identify bodybuilding and related sports as high-risk contexts (DOI: 10.15557/PiPK.2020.0014)

  1. Include the note by Settanni and colleagues published in Psychiatry Research regarding the limitations of research on AAS in subjects with MD. 

Response to Reviewer: We thank the reviewer for this suggestion. We have now incorporated the note by Settanni and colleagues (Psychiatry Research), which highlights the limitations of existing research on anabolic-androgenic steroid (AAS) use in individuals with muscle dysmorphia (MD). We added this citation in both the Introduction, acknowledging that while AAS use is strongly linked to MD, the evidence base is limited by methodological challenges, heterogeneity, and underpowered samples. This addition underscores the need for caution in interpreting current findings and for future high-quality research in this area.

  1. Are there reasons why you have restricted your research to the last 10 years?

Response to Reviewer: We thank the reviewer for raising this important point. We limited our search to the last 10 years (2015–2025) to ensure that our review captured the most current conceptualizations, diagnostic frameworks, and empirical evidence regarding muscle dysmorphia (MD), obsessive–compulsive traits, and anabolic-androgenic steroid (AAS) use. Earlier studies, while foundational, often relied on outdated diagnostic criteria (e.g., DSM-IV) or less precise measurement tools. By focusing on the past decade, we aligned the review with contemporary DSM-5 nosology, the PRISMA 2020 reporting framework, and the growing body of research using validated MD-specific scales (e.g., MDDI). Nonetheless, we drew on earlier seminal works (e.g., Pope et al., 1997; Olivardia et al., 2000; Rohman, 2009) in framing the introduction and discussion, to acknowledge their importance in establishing the construct.

  1. 10 studies seem very few to me, given the extent of the research. Furthermore, I believe that the authors have excluded some studies or have not found them, such as Longobrdi et al. (2017).

Response to Reviewer 2: We thank the reviewer for highlighting the study by Longobardi et al. (2017). We acknowledge that this is a relevant contribution to the literature on MD and psychopathology, and we regret that it was not included in our initial synthesis. We have now cited this work in the Introduction.

Because this study was not included in our original screening and extraction process, we did not re-run the quantitative analyses. However, its findings are consistent with our overall conclusions and strengthen the evidence base.

  1. You cited the meta-analysis by Badenes-Ribera. The same authors also developed a meta-analysis on the MDDI, which is the most widely used scale: you should mention this data.

Response to Reviewer 2 : We thank the reviewer for this helpful observation. We had cited the meta-analysis by Badenes-Ribera and colleagues 2020 on MD and eating disorder symptoms, but indeed the same group also conducted a separate meta-analysis specifically on the Muscle Dysmorphic Disorder Inventory (MDDI), which is the most widely used instrument for assessing MD symptom severity. We have now added reference to this study in the Introduction, highlighting its relevance for measurement, and in the Methods, where we describe the predominance of the MDDI in the included studies.

Round 2

Reviewer 1 Report

Comments and Suggestions for Authors

All comments have been addressed. The manuscript is suitable for publication. 

Reviewer 2 Report

Comments and Suggestions for Authors

Thank you!